# Oyster Shell Powder, Zeolite and Red Mud as Binders for Immobilising Toxic Metals in Fine Granular Contaminated Soils (from Industrial Zones in South Korea)

**DOI:** 10.3390/ijerph18052530

**Published:** 2021-03-04

**Authors:** Cecilia Torres-Quiroz, Janith Dissanayake, Junboum Park

**Affiliations:** 1Department of Civil and Environmental Engineering, Seoul National University, 1 Gwanak-ro, Gwanak-gu, Seoul 08826, Korea; ctorresq@snu.ac.kr (C.T.-Q.); janith1993@snu.ac.kr (J.D.); 2Institute of Construction and Environmental Engineering, Seoul National University, 1 Gwanak-ro, Gwanak-gu, Seoul 08826, Korea

**Keywords:** low-cost absorbent, stabilization and solidification, toxic metal, chemical stabilization, leaching test, TCLP, remediation

## Abstract

Low-cost absorbent materials have elicited the attention of researchers as binders for the stabilisation/solidification technique. As, there is a no comprehensive study, the authors of this paper investigated the performance of Oyster shell powder (OS), zeolite (Z), and red mud (RM) in stabilising heavy metals in three types of heavy metal-contaminated soils by using toxicity characteristic leaching procedure (TCLP). Samples were collected from surroundings of an abandoned metal mine site and from military service zone. Furthermore, a Pb-contaminated soil was artificially prepared to evaluate each binder (100× regulatory level for Pb). OS bound approximately 82% of Pb and 78% of Cu in real cases scenario. While Z was highly effective in stabilizing Pb in highly polluted artificial soil (>50% of Pb) at lower dosages than OS and RM, it was not effective in stabilising those metals in the soils obtained from the contaminated sites. RM did not perform consistently stabilising toxic metals in soils from contaminated sites, but it demonstrated a remarkable Pb-immobilisation under dosages over than 5% in the artificial soil. Further, authors observed that OS removal efficiency reached up to 94% after 10 days. The results suggest that OS is the best low-cost adsorbent material to stabilize soils contaminated with toxic metals considered in the study.

## 1. Introduction

Soil contaminated with toxic metals is a serious environmental issue worldwide [1,2]. Toxic metals exhibit the potential of affecting the soil–food chain, impairing soil fertility [3,4,5] and the quality of drinking water [6]. Once they enter the food chain, toxic metals can trigger cell mutation, possibly causing cancer [5,7,8]. Amongst these metals, Pb, Cd, Cu, Zn and Ni have elicited considerable concern because they are leached from tailings and discharged directly into adjacent streams and agricultural lands [9]. Numerous remediation techniques are available for remediating soil contaminated with toxic metals. They include surface capping [10], encapsulation [11,12], landfilling, soil flushing [13], soil washing [14,15], electrokinetic extraction [16,17,18], stabilisation/solidification (S/S) [19,20,21], vitrification [22,23], phytoremediation [24,25] and bioremediation. Amongst these techniques, S/S has attracted the attention of many researchers due to its low-cost application whilst preserving the long-term stability of the stabilised soil [26,27]. This technique involves the addition of binding materials (binders) to contaminated soil to stabilise and immobilise contaminants [28,29,30] via the chemical fixation of pollutants. This process is achieved through the interactions between the hydration products of binders and contaminants or the physical adsorption of contaminants [27,31]. The addition of lime, cement and other cementitious binders to soil has demonstrated excellent performance in treating soil contaminated with toxic metals [32]. However, stabilising wide areas of polluted soil by using such cementitious materials is economically infeasible due to the cost associated with such binders. To address this issue, many researchers have investigated nature-based materials, such as chitosan [33], zeolite [34], compost [35], hydroxyapatite [32] and waste products from certain industries, including fly ash [29,36], oyster shell powder [37,38], red mud [39,40] and coal, for their potential use as binders. At present, these materials have gained popularity as binders because of their local availability and low cost [26,30]. Oyster shell have been investigated for its interactions with toxic metal ions in an aqueous medium [38,41,42]. However, only a few researchers have investigated its application to soil [30,43,44,45]. The sorption characteristics and cation exchange capacity of zeolite has been studied extensively for its potential in removing toxic metals from water [34,46,47,48,49]. Similar to oyster shell, zeolite has not been investigated in terms of its use in remediating soil, although a few studies, such as those of Kwon et al. [50] and Wen and Zeng [51], investigated this subject. Red mud; Bauxite residue produced during Alumina production [41] is a relatively new material that has gained popularity in remediating soil [39]; however, its application has been limited due to the health risks associated with its use [52,53]. Some researchers have demonstrated that oyster shell, zeolite and red mud can be used as binders due to their sorption characteristics. Nevertheless, the literature on assessing their performance under various environmental settings is scarce. The characteristics of the medium, such as pH, Fe_2_O_3_ content and redox potential, can influence the binding process of these materials [21,54]. Water percolation enhances metal mobilisation in already stabilised soil because the H^+^ ions in acidic water displace the cations from their binding sites and reduce cation exchange capacity in accordance with Zheng et al. [31]. Thus, investigating the performance of these binders in leaching is crucial. 

As granular soils have a poor capacity to retain these toxic metals, the binders could be used to stabilize soils quickly before the metals leach into ground water levels. Therefore, the current study investigated the performance of oyster shell, zeolite and red mud in stabilising two actual samples of sandy soil contaminated with toxic metals at the laboratory scale, with the pollution levels close to permissible limits. Furthermore, the authors examined the performance of these binders in stabilising soil with high Pb concentration by applying binders to a handmade contaminated soil. Pb is frequently found in high concentrations in contaminated sites worldwide [55], and it could be immobilised using the binders considered in the current study. Therefore, the authors performed leaching tests by using deionised (DI) water and the toxicity leaching characteristic procedure (TCLP) to observe the performance of oyster shell, zeolite and red mud as binder stabilisers for remediating the three types of soil. Furthermore, the influences of contact time and soil pH on the performance of these binders were investigated. 

## 2. Materials and Methods

### 2.1. Contaminated Soil Samples

Soil samples were collected from two sites contaminated with toxic metals, namely, soil from surroundings of an abandoned metal mine site that was classified as silty sand (SM) with a fine content of 22% (denoted as “silty sand”) and soil from a military service area that was classified as well-graded sand (SW) with a fine content lower than 5% (denoted as “sandy soil”). The sites are respectively located at 62, 26 beon-gil, Gaegeumonjeong-ro, Gaegeum-dong, Busan, and San 65-1, Jangan-ri, Jangan-eup, Gijang-gun, Busan, South Korea. All the soil samples were air-dried and passed through a 2 mm mesh prior to preliminary analysis and experiments. The preliminary tests indicated that the soil samples had toxic metal concentrations closer to the regulation level. Therefore, handmade contaminated sand soil (HCS) was prepared at 100× the maximum permissible level for Pb, i.e., 3 mg/L, in accordance with the Korean regulation level to evaluate binder performance under high Pb concentrations. 

In the absence of standard methods for preparing contaminated soils, HCS was prepared following an approach similar to that of Martini and Shang [56]. HCS was made by mixing 4 kg of residual weathered soil (passed through a 2 mm mesh) classified as well-graded sand with Pb solution. Firstly, the weathered soil was sieved and analysed for the presence of toxic metals to determine the initial concentrations of the considered metals. Secondly, 67 g of PbCl_2_ was dissolved in 2 L of DI water. The solution was introduced into the soil and thoroughly mixed using an electric mixer until homogeneous slurry was achieved. The slurry was kept at room temperature for 2 days and then dried at 100 °C for 24 h. Lastly, the spiked soil mixture was homogenised by diagonally flipping it 3–5 times on a plastic sheet. The soil samples were kept at room temperature until the experiment was performed. The major chemical compounds observed in each soil are listed in Appendix A

### 2.2. Binders

Oyster shell powder (Jisan Industrial Co., Ltd., Busan, Korea with 89.3% CaCO_3_), natural zeolite (Silicon dioxide 61%, Mordenite 23.8%, Heulandite-Ca 15.4%; from Geumnong Industrial Co., Ltd., Pohang, Korea) and Red mud (Sanha E&C Co., Ltd., Gyeonggi, Korea) were evaluated via energy-dispersive X-ray spectroscopy (EDS) to determine the chemical composition of these binders. The major chemical compounds observed in each binder are listed in Appendix A. This chemical composition coincides with the findings of Shin, et al. [41], Xu, et al. [42] and Lu, et al. [38]. Moreover, the structure of each binder (i.e., surface and porosity) was observed in macroscale via scanning electron microscopy (SEM) (Figure 1). All the binders were washed with DI water, dried, and sieved (through a 0.15 mm mesh). Besides, the initial pH of Red Mud was lowered to 8.5–9 prior to by using 1 M HCL. 

### 2.3. Binder Performance Evaluation Method

In the current study, leaching with DI water following a procedure similar to that used in HJ-557-2010 [57] accelerated the mixing of soil treated with a binder. The liquid: solid ratio (L/S) and mixing time presented in HJ-557-2010 was adjusted to obtain a homogeneous mixture of soil and binder. Firstly, a batch leaching tests with the modified DI water leaching procedure was performed on the control samples (i.e., contaminated soil without binder). 

The experiment procedure is described as follows. All the samples were tested at least in duplicate.
The test was initiated by taking samples of 50 g of air-dried soil as the control and measuring the initial toxic metal concentrations (Pb, Cu, Zn, Cd and Ni). Then, 50 g of stabilised soil under different binder dosages per total weight (1, 3, 5, 7 and 10 wt%) was placed in a 250 mL glass flask and agitated for 2 h at 150 rpm with DI water at an L/S of 3.The supernatant fluid from the previous step was extracted 8 h after Step 1. For silty sand soil, however, additional extractions were performed at 12, 24 and 36 h after the first extraction to evaluate the effect of contact time with the binder on soil. After the supernatant fluid was extracted, it was filtered using a 0.45 μm membrane filter and then collocated in a 14 mL tube for toxic metal (Pb, Cu, Zn, Cd and Ni) concentration measurement via inductively coupled plasma optical emission spectroscopy (ICP-OES). Furthermore, pH was measured using a Thermo Scientific Orion 5-Star Plus Portable pH/ORP/ISE/Conductivity/DO Multiparameter Meter Model Number: PH3642-2(Beverly, MA, USA) as presented in Appendix A.The control samples (without binder) and stabilised soil (after Step 2, solid phase) were placed in an oven and dried at 60 °C for 24 h.TCLP test was conducted on all the soil samples obtained after Step 3.

TCLP test was performed to measure toxic metal concentration in accordance with the U.S. EPA Method 1311 [58,59] because CH_3_COOH, as an extract reagent, achieves better harmonisation during the laboratory testing of leaching compared with other reagents, such as EDTA [60]. Furthermore, acetic acid was used as the reagent because it represents a scenario in which organic acids are found in leachates from landfills [55].The steps for the TCLP test are described as follows.
5.A 2 g sample (from Step 3) was placed in small tubes that contained 40 mL of the extract solution (L/S = 20). Extract solution type depends on the pH of the medium (previously measured in Step 2).6.After mixing thoroughly using a rotary tumbler at 30 ± 2 rpm for 18 h, the samples were allowed to settle for 12 h. Then, the supernatant fluid was extracted and sieved using a 0.45 μm membrane filter and collocated in tubes with a 14 mL capacity to measure toxic metal concentration via ICP-OES. For the HCS treated with oyster shell, additional extractions were performed after 1 day and 10 days of mixing to evaluate the effect of contact time.7.Lastly, the pH of the leachate was measured and reported in Appendix A. A summary of the experiment procedure is presented as Figure 2.

### 2.4. Measurement of Initial Toxic Metal Concentrations

The physiochemical properties and initial toxic metal concentrations of all the soil samples are presented in Table 1. The concentrations measured in the leachate (mg/L) via ICP-OES was converted into mg/kg by using Equation (1) in the Korean standard procedure ES 07400.2c [61]. The same method was described [62,63,64].
(1)$$C_{\left(\frac{\mathrm{mg}}{\mathrm{kg}}\right)}=\frac{\left(C_{1}-C_{0}\right)}{W_{d}}\times f\times V$$
where
*C*_1_: metal concentration of the analytical specimens obtained from the calibration curves (mg/L),*C*_0_: metal concentration of the blank solution obtained from the calibration curve (mg/L),*f*: dilution rate,*V*: volume of the specimen container and*W_d_*: dry weight of the soil specimen

The toxic metal concentrations prior to the addition of a binder converted as a fraction of mass (mg/kg) are provided in Table 1.

**Table 1 ijerph-18-02530-t001:** Physiochemical properties of silty sand soil, sandy soil and handmade HCS; toxic metal concentrations in leachates before binder application.

Source	USCS ^1^	pH	Extraction Method	Extract Fluid ^2^	Initial Concentrations				
					Pb	Cu	Zn	Cd	Ni
			DI (mg L^−1^)	DI	0.015	0.110	0.045	0.002	-
Case I: Mine area	SM	8.1	TCLP (mg L^−1^)	I	0.639	3.954	102.784	0.316	0.432
			TCLP (mg Kg^−1^)		12.780	79.080	2055.680	6.320	8.640
			DI (mg L^−1^)	DI	0.550	0.210	0.450	0.003	-
Case II: Military area	SW	6.7	TCLP (mg L^−1^)	I	0.079	2.235	10.053	0.046	-
			TCLP (mg Kg^−1^)		1.580	44.700	201.060	0.920	-
			DI (mg L^−1^)	DI	301.657	0.440	-	-	
Case III: HCS	SW	4.9	TCLP (mg L^−1^)	II	159.802	0.444	-	-	-
			TCLP (mg kg^−1^)		3196.04	8.880	-	-	-

^1^ Unified Soil Classification System—USCS; ^2^ USEPA Method 1311.

## 3. Results

### 3.1. Effects of Binder and Dosage

#### 3.1.1. Case I: Silty Sand Soil from an Abandoned Metal Mine Site

The silty sand soil had an initial pH between 7.97 and 8.29 due to the presence of CaO (Table 1). When mixed with different dosages of oyster shell, zeolite and red mud, the pH of the soil changed to 7.87–8.18, 7.60–7.81 and 7.89–9.12, respectively (Appendix A). The dosages of oyster shell and zeolite exerted no significant effect on the pH of this soil, whereas red mud dosage had a significant effect on pH. 

We observed that this soil had an initial Cu concentration (Table 1) that exceeded the South Korean regulation value for leachate and the World Health Organization (WHO), Australian and Canadian guideline values for soil (Appendix A). Moreover, the silty sand soil contained Cd and Zn concentrations beyond the recommended guideline values. Other toxic metal concentrations were below the guideline limits. After adding 5 wt% of oyster shell, the concentration of Cu in the leachate was reduced from 3.954 mg/L to 0.937 mg/L (i.e., a reduction of 76%, Appendix A), which is below the Korean regulation (<3 mg/L) and the limits stipulated by WHO (Appendix A) (Figure 3 and Appendix A). Zn was reduced from 102.784 mg/L to 68.657 mg/L, which is also under relevant limits (Appendix A). Moreover, Cd concentration in the leachate decreased from 0.316 mg/L to 0.182 mg/L, satisfying the condition for toxic metal presence in wastewater [65]. Although the initial Pb and Ni concentrations were below the regulatory values, their concentrations were also reduced with increasing oyster shell dosage (Figure 3 and Appendix A). In the case of zeolite, a dosage of 1 wt% reduced Cu and Pb concentrations by 50%; thereafter, binder dose exerted no further effect (Figure 3 and Appendix A). This phenomenon was also observed for Zn and Ni with a dosage of over 3 wt%. Furthermore, zeolite appeared ineffective in binding Cd in soil at any dosage. Although the leachates from the soil were stabilised with zeolite at concentrations below the relevant guidelines (except for Cd), the reduction was considerably lower compared with that of oyster shell. However, the leachate from the silty sand soil mixed with red mud presented a higher amount of Cd than the initial concentration, suggesting the poor adsorption of Cd by red mud (Figure 3 and Appendix A). A dosage of 3 wt% was effective for stabilising Cu and Zn, whilst increasing the dosage from 3 wt% to 5 wt% yielded no significant benefit (*p* > 0.05). Similar to oyster shell, the performance of red mud in binding Pb and Ni increased with dosage. 

#### 3.1.2. Case II: Sandy Soil from a Military Service Area

Meanwhile, the sandy soil had an initial pH between 6.55 and 6.85. When mixed with oyster shell, zeolite and red mud, pH changed to 7.59–8.04, 6.25–6.58 and 7.90–9.48, respectively (Appendix A). The addition of oyster shell and red mud increased the pH of the medium, whereas the addition of zeolite reduced the pH of the medium. In the case of red mud, pH increase was significant. 

Initially, the sandy soil was slightly contaminated with Zn (201.060 mg/kg) on the basis of the Korean, Canadian, Australian (<200 mg/kg) [27,66] and WHO (<50 mg/kg) [67] regulations, as presented in Appendix A. Other toxic metal concentrations (Figure 3 and Appendix A) were under the guideline values. Zn concentrations in the leachate were below the maximum permissible level for all the soil samples stabilised with 5 wt% dosage of any of the binders considered in this study. With a 5 wt% dosage of oyster shell, zeolite and red mud, Zn immobilisation rates of 50%, 37% and 32% were observed, respectively, as shown in Appendix A. In the case of zeolite, increasing the dosage after 1 wt% did not improve the immobilisation of toxic metals. The addition of high dosages of red mud increased Pb concentration by a factor close to two. Although Cu was below the relevant guideline values, our findings implied that applying oyster shell, zeolite and red mud immobilised 86%, 27% and 37% of Cu, respectively. Overall, oyster shell exhibited the highest immobilisation rate for all the toxic metals considered in this study. 

#### 3.1.3. Case III: Handmade Contaminated Soil (HCS)

HCS, which is also sandy soil, had an initial pH between 4.75 and 5.05. After mixing with oyster shell, zeolite and red mud, its pH changed to 6.91–7.51, 4.42–5.18 and 6.23–9.63, respectively (Appendix A). Compared with the initial pH values, the addition of oyster shell and red mud increased the pH of HCS. By contrast, the addition of zeolite at dosages lower than 5% decreased pH slightly whilst dosages over 5% increased pH marginally.

The leachate with DI water presented an initial Pb concentration of 301.65 mg/L. Conversely, the leaching test with TCLP reported a mean Pb concentration of 159.802 mg/L as the initial concentration of HCS. This value is lower than the concentrations obtained by leaching with DI water. This finding can be attributed to the aging effect (short time). The results suggested that Pb concentration can be reduced by 62% by adding 5 wt% of zeolite. However, the final concentration (113.825 mg/L) was still higher than Korean and international regulations (<5 mg/L, Appendix A). Meanwhile, Pb concentration in the leachate was reduced to 269.942 mg/L (11% of Pb was immobilised) after administering oyster shell and 177.637 mg/L (41% of Pb was immobilised) after administering red mud at 5 wt% dosage. However, when dosage was increased to 10 wt%, the immobilisation rate via oyster shell addition significantly improved. By contrast, such an improvement in binder performance was not observed with zeolite when its dosage was increased. After adding 10 wt% of oyster shell, zeolite and red mud, Pb immobilisation rates of 53%, 64% and 59%, respectively, were observed (Figure 3 and Appendix A). 

The initial Cu concentration was lower than the maximum permissible levels. However, after adding 5 wt% of oyster shell, zeolite and red mud, 29%, 55% and 53% of Cu, respectively, were immobilised. When dosage was increased to 10 wt%, the immobilisation rates increased to 61%, 63% and 61%, respectively. Similar to the observations for Pb, the increment in immobilisation by oyster shell was significant when dosage was increased (Appendix A).

### 3.2. Effect of Contact Time

When silty sand soil with moderate toxic metal pollution level was mixed with oyster shell, Pb concentration decreased immediately, immobilising up to 70% of Pb; thereafter, it did not improve with increasing contact time (Appendix A). A similar phenomenon was observed for Zn and Cu. The immobilisation rates of Zn and Cu in this soil sample stabilised with zeolite and red mud improved with longer contact time. Zeolite and red mud exhibited constant performance in stabilising Pb at different contact times. 

For HCS soil stabilised with oyster shell, the high Pb concentration in the leachate was reduced to 38.769 mg/L (87% of Pb was immobilised) after 1 day of contact time and 94% of Pb was immobilised after 10 days at dosages above 5% (Figure 4 and Appendix A). However, these values were still above the regulation level. When HCS was treated with oyster shell at 10 wt%, the Pb concentration in the leachate collected after 1 day was 3.67 mg/L, which is slightly above the Korean regulation but under the maximum value of 5 mg/L stipulated by U.S. EPA. At short contact times (e.g., 12 h), oyster shell with dosages up to 5% immobilised less than 10% of Pb. After 1 day, oyster shell apparently reached maximum Cu and Pb stabilisation because the authors did not observe any significant improvement in immobilisation.

## 4. Discussion

### 4.1. Oyster Shell Powder

Oyster shell powder demonstrated the best performance in binding nearly all the considered toxic metals, particularly Pb and Cu, for the silty sand and sandy soil samples. Furthermore, our observations suggested that the preference for sorption of oyster shell was in the following order: Pb^2+^ > Cu^2+^ > Zn^2+^ > Cd^2+^ > Ni^2+^. This order is similar to the observations of Shin et al. [41] in their kinetic model. In addition, when HCS was treated with oyster shell, all the samples had a pH above 7 (alkaline). CaCO_3_ and CaO in oyster shell were dissolved in water to produce hydroxyl ions (OH^−^), increasing the pH of the medium (CaCO_3_ + H_2_O → Ca^2+^ + CO_3_^2−^; CO_3_^2−^ + H_2_O → HCO_3_ + OH^−^) [68]. This alkaline condition can promote the precipitation of metals as metal hydroxides [M^n+^ + n(OH)^−^ → M(OH)_n_, where M denotes metal] [68,69], and can be linked to the reduction of toxic metals in leachate. Furthermore, the SEM–EDS analysis (Figure 5) showed that oyster shell exhibited high adsorption capacity towards Pb compared with the other binders considered in this study (SEM analysis results for other binder doses are presented as Appendix A). This finding can be attributed to ion exchange capacity [30]. The two aforementioned phenomena can justify the significant reduction (*p* < 0.05) in Pb with increased oyster shell dosage and contact time. The number of sorption sites and reactive hydroxide ions increased with oyster shell dosage, significantly reducing toxic metal concentrations (*p* < 0.05), as illustrated in Appendix A (descriptive statistics pertaining to the concentrations of toxic metals at different binder dosages are presented in Appendix A). In the case of HCS treated with 5% oyster shell dosage, only 30% of Cu and 10% of Pb were immobilised. When dosage was increased to 10%, sorption sites and metal precipitates were consequently increased and immobilisation rate reached up to 60% for Cu and 55% for Pb. 

Furthermore, the results of Pb and Cu concentrations in the HCS leachate stabilised with oyster shell over time. Figure 4 shows that better immobilisation of these toxic metals can be achieved with increasing contact time. The effect of contact time seemed insignificant for dosages over 5% and the change in immobilisation rate was insignificant (*p* > 0.05) for contact times beyond 1 day. This finding was also observed in silty sand soil, as illustrated in Appendix A. The preceding results corroborate the findings of Xu, et al. [42] and Desta [63], who observed the roles played by the adsorption characteristics and ion exchange capacity of binders and the precipitation of ions in achieving stabilisation over time. Considering that the immobilisation rate was maintained without significant variation, oyster shell can be used to stabilise a wide range of soil types; silty sand, sandy soil and HCS. Although oyster shell exhibited good performance in binding toxic metals, the treatment of soil that is highly polluted with Pb is recommended only for industrial areas where the percolation of water can be controlled. In addition, oyster shell contains sodium, which may be harmful to flora in excessive dosages. However, appropriate doses of oyster shell can be used as soil amendment for agricultural soil [69,70].

### 4.2. Zeolite

In the current study, zeolite-treated soil had a neutral pH of approximately 6.5 and zeolite did not drastically change the pH of the medium. A slight increase in pH was observed when zeolite dosage was increased. Such pH conditions are beneficial for stabilisation by zeolite because the major binding mechanisms of zeolite are adsorption and cation exchange [68,69]. This phenomenon can be observed in HCS soil (a slightly acidic soil) wherein zeolite achieved the best immobilisation compared with the other binders at a contact time of 12 h. Moreover, given that the immobilisation rate does not improve with dosage, we can argue that ion exchange is the primary binding mechanism of zeolite in our study. The surface of zeolite is negatively charged through the isomorphous replacement of Al^+^ by Al^3+^. This negative charge can be balanced by exchangeable cations, such as Na, K and Ca. These exchangeable cations are used in ion exchange with metals ions, such as Pb, Cd, Zn and Cu [34,50,71]. The number of exchangeable metal ions in soil did not change with increased dosage, and ion exchange may not occur because the activity of metal ions in the medium became considerably low. When exchangeable Pb ions are high in HCS, even a low dosage of 1% zeolite can bind 50% of Pb because of this high ion exchange capacity. 

The leachates obtained for silty sand and sandy soil with DI water indicated that the addition of zeolite significantly reduced the concentrations of Pb and Cu (Appendix A); however, the immobilisation rate was lower compared with that of oyster shell. This result may be attributed to the low activity of the ions present in the soil. Furthermore, the final concentrations of toxic metals in the leachates of silty sand and sandy soil at 5 wt% were in the following order: Pb < Cu < Zn. This finding corroborates zeolite’s selectivity for cation exchange, i.e., Pb^2+^ > Cu^2+^ > Zn^2+^ [26]. Zeolite can be applied to soil near military bases or industries that is contaminated with a high amount of Pb or to acidic soil. In the current study, the authors only observed the performance after a contact time of 12 h. Therefore, observing the long-term performance of zeolite is necessary because ion exchange is a reversible process in zeolite.

### 4.3. Red Mud

Red mud has an alkaline nature because of the presence of NaOH, a strong base used in producing alumina [72]. The high alkalinity of red mud can increase pH drastically as we observed in this study. As shown in Figure 3, red mud is effective for highly contaminated soil, such as HCS. Shin, et al. [41] reported that red mud has a larger surface area than oyster shell and zeolite, and this characteristic improves the adsorption capacity (primary components: silica, alumina and Fe_2_O_3_) and ion exchange capacity of this material. This characteristic may be the reason for the previous observation. Furthermore, red mud can be used to precipitate soluble toxic metals in their hydroxide form. Therefore, a higher red mud dosage will result in higher stabilisation of Pb, as observed in Figure 3, particularly for HCS soil. In sandy soil, a higher Cd concentration was observed after treatment, suggesting that Cd is adsorbed poorly in a competitive environment because red mud can exchange Cd ions in soil with Al compounds. Red mud was ineffective for soil with low contamination, such as sandy soil. In fact, the addition of red mud was counterproductive, particularly in the case of Pb. This finding may be ascribed to the increase in pH and the dissolved organic carbon in soil pore water [73,74]. Dissolved organic carbon can enhance the leaching of As, Cu and Ni from red mud when the latter comes in contact with organic-rich media [52], such as the silty sand soil evaluated in this study.

Considering the presence of water-soluble Al concentrations in red mud, biologically available Al can be released into the surrounding environment; in its acutely toxic form, i.e., [Al(OH)4]^−^, Al can pose considerable environmental and health hazards [73]. Therefore, the application of red mud to fertilised soil is not recommended.

## 5. Conclusions

The S/S technique aims to immobilise contaminants by converting them into a less soluble form (chemical stabilisation) and encapsulating them by creating a durable matrix (solidification), as observed through pH measurements after mixing the binder. From the results, binder performance changed depending on the type and level of toxic metal concentration (HCS > silty sand soil > sandy soil), and the pH of the final medium. These factors are associated with the solubility and mobilisation of toxic metals. When the medium was alkaline, better binding was observed amongst all the binders, emphasising the role of OH^−^ ions in aiding the precipitation of toxic metals.

Oyster shell demonstrated the best performance in binding Pb and Cu in the silty sand soil and sandy soil (including Cd and Ni), and its effect was immediately observable after adding 3% of the binder to soil samples with low-to-medium contamination levels. Moreover, oyster shell proved to be a good binder even for soil with an extremely high Pb concentration (i.e., HCS). However, higher dosages (>5%) and longer contact times (>1 day) are required to achieve the desired immobilisation rates. Therefore, OS can be used to stabilise soils contaminated with of Pb and Cu.

Zeolite is a good alternative binder for highly contaminated soils (even under acidic condition) because of its cation exchange capacity with toxic metals and sorption properties. However, no significant improvement in binding performance can be achieved with doses above 3%. In this experiment, the leaching agent was in contact for only 12 h; thus, observing the performance of zeolite in a long-term setting is recommended. Red mud can be used alternatively to Zeolite. However, caution should be taken during its application because of the risk of releasing other metalloids. However, the performance of both Zeolite and Red mud is not reliable compared to Oyster shell.

The infiltration of acid rain may decrease soil pH, leading to the mobilisation of bound toxic metals. Therefore, investigating the effect of pH and conducting a column percolation test prior to recommending any applications are recommended because most of the binders demonstrate better binding properties under alkaline conditions, particularly at low dosages.

For future studies, the authors suggest investigating whether the surface area of binders can be increased by subjecting them to high temperatures. Furthermore, biotic redox reactions, which are important for controlling oxidation state, were not considered in the current study. Thus, how the mobilisation of toxic metals is affected by such reactions requires investigation.

## Figures and Tables

**Figure 1 ijerph-18-02530-f001:**
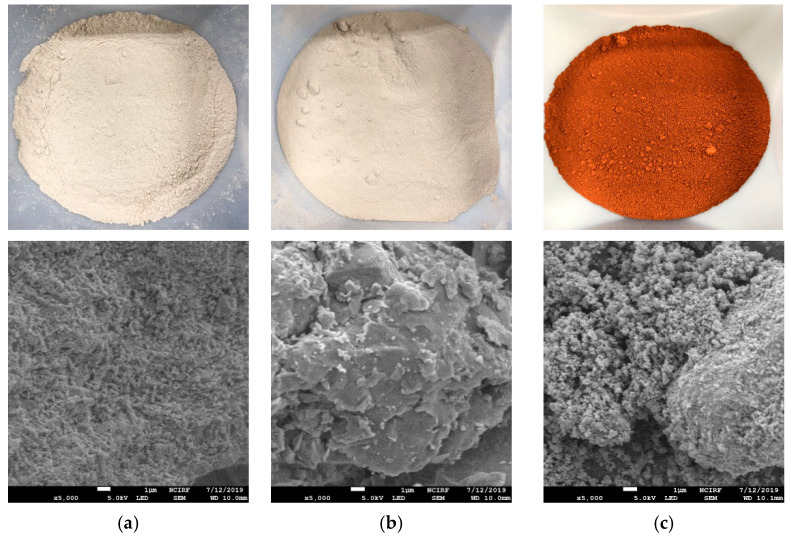
Materials texture and SEM analysis of the binder stabilisers considered in the study (5000×) (**a**) Oyster shell powder (OS); (**b**) Zeolite (Z) and (**c**) Red Mud (RM).

**Figure 2 ijerph-18-02530-f002:**
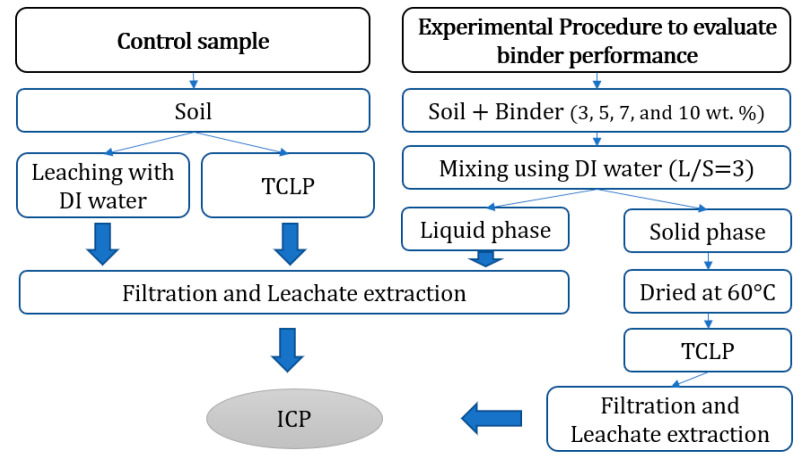
An overview of the experimental procedure for both the control sample (without binder) and soil after stabilized with the binder.

**Figure 3 ijerph-18-02530-f003:**
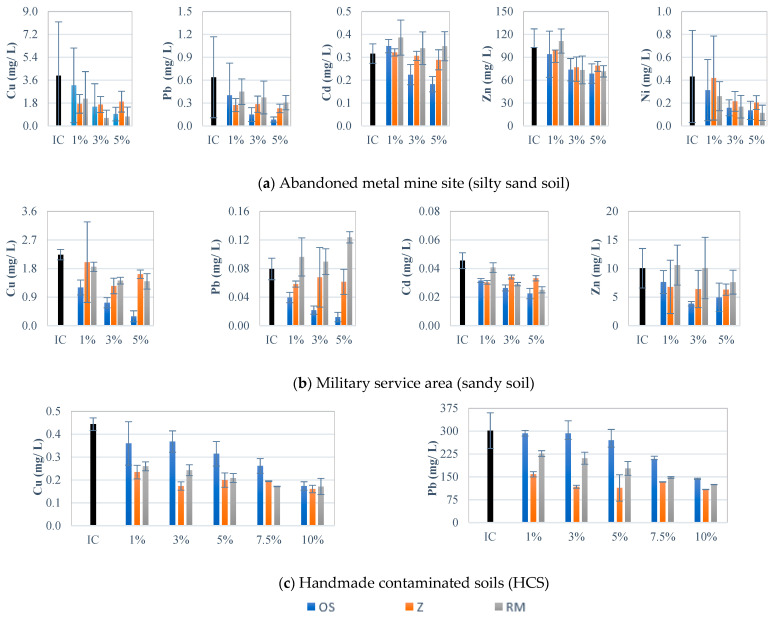
Toxic metal concentrations in mg/L (from the TCLP test results of the silty sand soil, sandy soil and HCS). (IC—Initial concentration; OS—Oyster shell; Z—zeolite; RM—Red mud). The number after the binder abbreviation represents mass percentage (e.g., Z3 means 3 wt% of zeolite). (Refer to Appendix A for the descriptive statistics of these values).

**Figure 4 ijerph-18-02530-f004:**
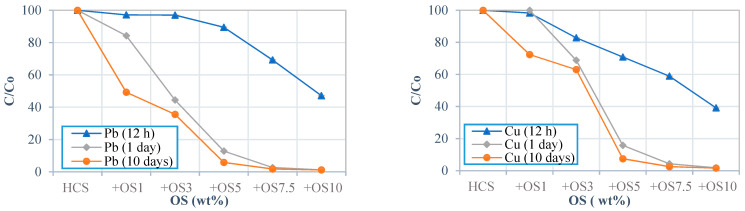
Pb and Cu concentrations in the HCS treated with oyster shell obtained through the TCLP test at different contact times (12 h, 1 day and 10 days). (The number after the binder abbreviation represents mass percentage (e.g., OS3 means 3 wt% of oyster shell). (Refer to Appendix A).

**Figure 5 ijerph-18-02530-f005:**
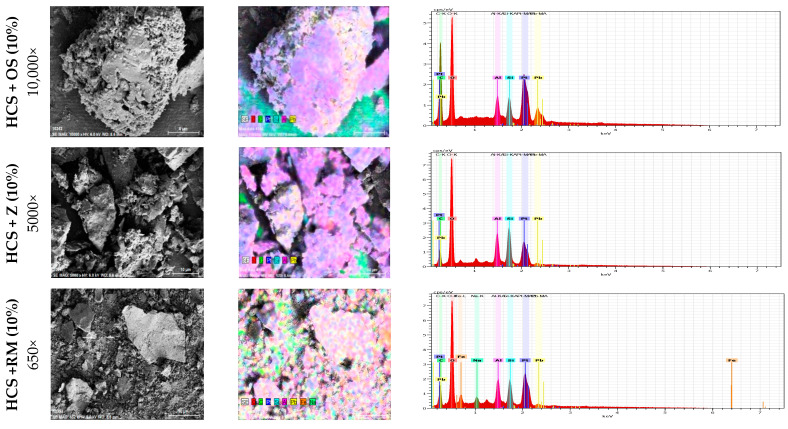
SEM–EDS analysis result for HCS treated with 10 wt% oyster shell, zeolite, and red mud.

## Data Availability

Data can be made available upon request.

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
