# Peer review of "Oyster Shell Powder, Zeolite and Red Mud as Binders for Immobilising Toxic Metals in Fine Granular Contaminated Soils (from Industrial Zones in South Korea)"

_ijerph, 2021, doi:10.3390/ijerph18052530_

Round 1
Reviewer 1 Report
The article presents an interesting topic concerning the immobilisation of heavy metals. The authors presented a lot of research results, which caused a mess in the presented article
Objective of the work:
There is no explanation why the authors stabilized the soil with lead?
Materials and methods:
- no description of the doses used
- 126: The experiment procedure is described as follows - it's better to present it on a diagram
- Table 1 - wrong table title- unnecessary values of TCLP metal concentration in mg / l
- „measure toxic metal concentration” - what exactly?
Results
-„The dosages of oyster shell and zeolite exerted no significant effect on the pH of this soil, whereas red mud dosage had a significant effect (p < 0.05) on pH” – How to find statistically significant?
- Figure 3- Why only Pb and Cu concentrations are described?
- Figure 4. SEM–EDS analysis result for HCS treated with 10 wt% oyster shell, zeolite, and red mud- why the authors chose only this analysis?
Dicussion:
„[41] reported that red mud has” – itl will be better Han et al. reported that …
In the discussion there are no comparisons to the work of other researchers.
Conclusion:
Conclusions should be detailed
Author Response
Thank you for your comments, please find our rebuttal in the attachment provided.

Reviewer 2 Report
The article is very interesting. The research is very ingenious. It is worth emphasizing that the research methodology is very thorough and described in detail. Such description of the methodology increases the credibility of the research. The table and drawing results included in the supplement are an excellent supplement. Reading an article with simultaneous
following the supplement explains any difficult parts of the topic. While my assessment of the article is without a doubt very positive, I have some small suggestions that could improve the manuscript.
- Zeolite was used in the research - it would be good to present which zeolite was used, or at least from which group. There are many zeolites in nature and they can differ significantly from each other.
- Please explain what red mud is.
Author Response

(The authors gave the same response as above.)

Reviewer 3 Report
The paper is very interesting but it needs some revisions before it can be accepted. The original of the binders is not clear. Authors write "Oyster shell, zeolite and red mud were obtained from domestic suppliers". What are domestic suppliers? From where the oysters come from? And the mud? From the same area as the authors? More details should be given in the Experimental part. Moreover, what type of zeolite is it? Also an image os shown where it seems that the oyster shell is in powder form? Was it crused? How? Was it supplied in that form? However it is needed to know what was made of it and if some other treatments or cleaning were made by the supplier.
Table 1 has no caption. It is only there the template text.
The first image shown it Figure 2 and the second is also Figure 2. The first should be numbered as Figure 1.
Some explanation is given for the oyster shell to be a better binder, but that is not stated in the Conclusions or Abstract and it should.
Author Response

(The authors gave the same response as above.)

Round 2
Reviewer 1 Report
I accepted your response